# Correcting modification-mediated errors in nanopore sequencing by nucleotide demodification and reference-based correction

Chien-Shun Chiou[1], Bo-Han Chen[1], You-Wun Wang[1], Nang-Ting Kuo[2], Chih-Hsiang Chang[2] & Yao-Ting Huang [2✉]

The accuracy of Oxford Nanopore Technology (ONT) sequencing has significantly improved thanks to new flowcells, sequencing kits, and basecalling algorithms. However, novel modification types untrained in the basecalling models can seriously reduce the quality. Here we reports a set of ONT-sequenced genomes with unexpected low quality due to novel modification types. Demodification by whole-genome amplification significantly improved the quality but lost the epigenome. We also developed a reference-based method, Modpolish, for correcting modification-mediated errors while retaining the epigenome when a sufficient number of closely-related genomes is publicly available (default: top 20 genomes with at least 95% identity). Modpolish not only significantly improved the quality of in-house sequenced genomes but also public datasets sequenced by R9.4 and R10.4 (simplex). Our results suggested that novel modifications are prone to ONT systematic errors. Nevertheless, these errors are correctable by nucleotide demodification or Modpolish without prior knowledge of modifications.

[1] Centers for Disease Control, Taichung, Taiwan. [2] Department of Computer Science and Information Engineering, National Chung Cheng University, Chiayi, Taiwan. ✉email: ythuang@cs.ccu.edu.tw

The Oxford Nanopore Technology (ONT) is a popular long-read sequencing platform that enables real-time sequencing for point-of-care medical applications, such as the diagnosis of acute infectious diseases within hospitals[1,2]. Despite its great potential and popularity, the accuracy of ONT was inferior to those of other platforms (e.g., Illumina and PacBio HiFi). Recently, the quality of ONT sequencing has significantly improved thanks to new flowcells (e.g., R10.4), sequencing kits (e.g., Kit 14), and basecalling algorithms (e.g., Bonito). For example, by using the R10.4 flowcells, near-perfect microbial genomes from isolates or metagenomes can be reconstructed by ONT-only sequencing without short-read polishing[3].

However, because R10.4 can only deliver the highest accuracy when using duplex reads of lower yield, most sequencing projects still adopt the simplex mode due to cost considerations. Consequently, postassembly genome polishing is still compulsory for removing ONT systematic errors. Systematic errors are recurrent basecalling errors at the same locus, which are not correctable by the consensus of read pileups (e.g., Racon)[4]. Homopolymer errors (i.e., indels) were the primary source of ONT systematic errors. Thanks to several machine-learning algorithms, these errors have been significantly reduced by read-based (e.g., Medaka) or reference-based (e.g., Homopolish) post-assembly polishing[5]. These algorithmic advances have produced high-quality ONT genomes sufficient for downstream analysis (e.g., >Q50)[3,6].

The ONT signals are ultra-sensitive to various modifications (e.g., 5mC, 6 mA) and therefore very suitable for epigenetic screening. To date, more than 17 and 160 modification types have been found in DNA and RNA, respectively, and the number is still growing (e.g., DNA adducts, N4-acetyldeoxycytosine)[7,8]. These modifications disturb the electrical current and result in systematic errors[9]. These modification-mediated errors are partially resolved by new flowcells and sequencing kits (e.g., R10.4 and Kit 14), which mainly reduce homopolymer errors. Furthermore, existing basecalling and polishing algorithms (e.g., Guppy and Medaka) were trained for capturing only a few modifications (e.g., 5mC, 5hmc, 6 mA). Consequently, the quality of ONT sequencing is unreliable when novel modification systems extensively edit the genome.

This paper presents a set of unexpected low-quality genomes due to extensive modification-induced errors. We show that the removal of modifications by whole-genome amplification (WGA) significantly improves the quality of all genomes. A novel computational method is developed for correcting these modification errors without the need for WGA.

## Results

**Unusual low-quality ONT genomes due to extensive modifications.** We sequenced 12 microbial strains of *Listeria monocytogenes* using Illumina and ONT R9.4 flowcells (~200–990Mbp, SUP model) (Fig. 1a, Supplementary Tables 1 and 2). The ONT reads were assembled into genomes with sequencing errors further polished by Medaka and Homopolish (Supplementary Table 3, see Methods). The Illumina and ONT read were hybrid assembled for evaluation purposes (Supplementary Table 4). When compared with the Illumina/ONT hybrid assemblies (Fig. 1b), seven ONT-only genomes exhibited high quality (HQ) ranging from Q47 to Q60 (e.g., R19-2905 and R20-0088). However, five isolates (R20-0026, R20-0030, R20-0127, R20-0148, and R20-0150) showed unexpectedly low quality (LQ) varying from Q26 to Q32. The accuracy of these five LQ genomes remained unimproved after replicated ONT sequencing. Further investigation of the five LQ genomes revealed excessive amounts of mismatch errors (1533–5670) compared with the seven HQ ones (0–40 mismatches) (Fig. 1c). Homopolymer errors (i.e., indels)

were not the source of inferior quality (7–306, Supplementary Table 5).

Manual inspection revealed that these mismatches were ONT basecalling errors uncorrected after genome polishing (Fig. 1d and Supplementary Fig. 1). As mismatch errors in ONT are mainly due to epigenetic modifications, we computed the frequency of well-known methylation in these isolates (see Method and Supplementary Table 6). In terms of 5-methylcytosine (5mC), the numbers of modified loci in the five LQ genomes (~240–340k) were not significantly higher than those in the HQ ones (210–345k, $P = 0.89$, Fig. 1e). Similarly, the numbers of N⁶-methyladenine (6 mA) modifications also showed no significant difference between the LQ and HQ groups (98–218k vs. 126–223k, $P = 0.34$). Because the numbers of mismatch errors in LQ genomes are significantly higher than those of HQ ones ($P = 0.005$), we suspected ONT basecalling algorithms failed to distinguish the novel modification types in the LQ isolates.

**High-quality ONT genomes by WGA demodification.** We removed the modifications in all microbial samples by WGA (Fig. 2a), which randomly amplifies the genome fragments without retaining any epigenetic modification (see Methods). The WGA-demodified samples were sequenced by ONT (R9.4), assembled into chromosomes, and compared with the Illumina/ONT hybrid genomes (Fig. 2a, Supplementary Tables 7 and 8). The five LQ genomes after WGA exhibited significantly higher quality than those without demodifications (e.g., Q26 to Q53 in R20-0026) (Fig. 2b, Supplementary Table 9). In particular, the amounts of mismatch errors significantly reduced after demodification (e.g., 5670 to 16 in R20-0026) (Fig. 2c). Consequently, the unexpected low quality of ONT was due to excessive modification-induced errors untrained in their basecalling model. The demodification by WGA can produce high-quality ONT genomes without the need for Illumina short reads.

However, while WGA successfully erased these modifications, the sequencing cost increased by two factors. First, WGA required a higher sequencing depth (~100×) for assembling a complete genome when compared with ordinary ONT sequencing (~30×) (Fig. 2d and Supplementary Figs. 2 and 3). It was due to the uneven amplification of WGA, which led to non-uniform sequencing depth and a fragmented assembly at moderate coverage. Second, the WGA-demodified samples may reduce the ONT yields. We observed the number of available/active pores could sometimes decrease quickly (e.g., less than 100 pores after 12 h) (Fig. 2e), which was possibly owing to the hyperbranched structure unresolved after WGA[10]. Consequently, the sequencing cost of WGA-demodified samples using ONT is much higher than ordinary sequencing.

**In silico correction of modification-mediated errors by Modpolish.** We developed a novel computational method (called Modpolish) for correcting these modification-mediated errors without WGA and prior knowledge of the modification systems. Modpolish identifies and corrects the modification-mediated errors by leveraging basecalling quality, basecalling consistency, and evolutionary conservation (Fig. 3a, see "Methods"). Briefly, because the ONT signals are disturbed by modifications, the basecalling quality is substantially lower than the modification-free loci (Supplementary Fig. 4). As such, the basecalled nucleotides are often inconsistent at the modified loci (Supplementary Fig. 5), yet these loci are within conservative motifs (Supplementary Fig. 6). In conjunction with the conservation degree measured by closely-related genomes, only the modified loci with

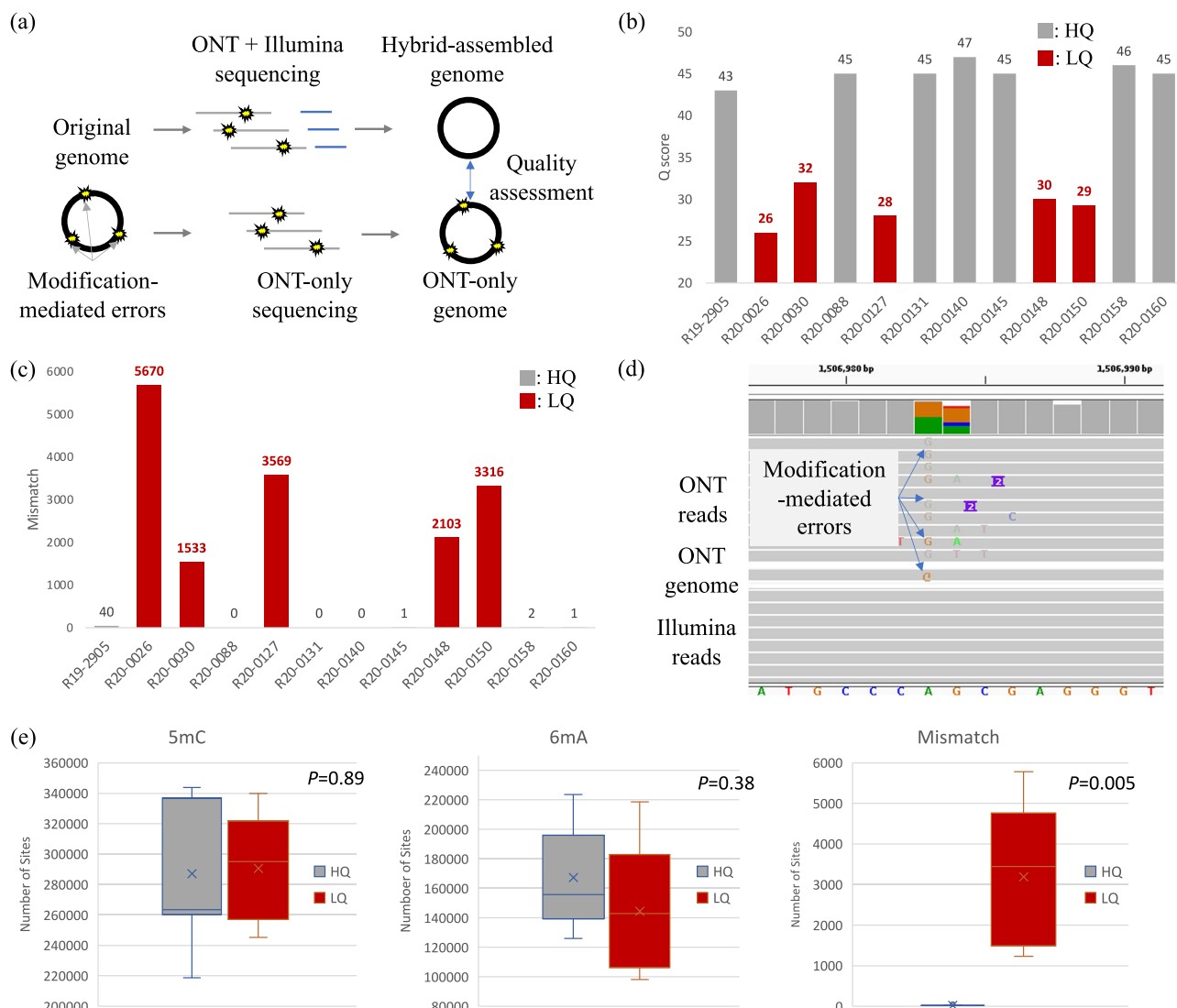

**Fig. 1 Quality comparison of 12 *Listeria* strains using ONT-only and ONT/Illumina hybrid sequencing. a** Workflow of ONT-only and ONT/Illumina hybrid assembly; **b** Q scores; **c** number of mismatches (red: LQ, gray: HQ); **d** comparison of ONT and Illumina reads by IGV; **e** numbers of 5mC, 6 mA, and mismatches between HQ/LQ strains ($n = 12$, red: LQ, gray: HQ). Error bars represent the minimum and maximum values.

ultra-high conservation will be corrected by Modpolish, avoiding false corrections of strain variations with high specificity.

We assessed the accuracy of Modpolish by comparing the quality of the ONT-only genomes (polished by Medaka) with those further polished by Modpolish. The results indicated that Modpolish significantly improved the quality of all LQ genomes from Q27–34 to Q60 (Fig. 3b, Supplementary Table 10). The number of mismatches also greatly decreased (e.g., from 5670 to 67 in R20-0026) (Fig. (3c). The numbers of mismatches in some HQ genomes were also reduced by Modpolish. For instance, the mismatches in the R19-2905 were erased from 40 to 6. Consequently, our results suggested that Modpolish made no false corrections on the HQ genomes (Supplementary Tables 11–13). The comparison of different basecaller versions and models (v4.0.14 vs. v6.3.4, HAC vs. SUP) indicated that these errors remain exist and Modpolish successfully erases most of them (Supplementary Fig. 7).

As the modification systems often involve anti-phage defense (e.g., R-M, BREX, DISARM)[11–13], we investigated the defending systems possessed by the HQ and LQ strains (Fig. 3d) (Supplementary Data 1). All the HQ genomes encompass at least one R-M system (e.g., Type I, II, or III), which is missing in all LQ

isolates. Instead, four LQ strains (i.e., R20-0030, R20-0127, R20-0148, R20-150) carry a novel methyltransferase-encoding *mza* defending system which is absent in all HQ genomes (Supplementary Fig. 8). Analysis of modification sites of the four *mza*-encoding LQ strains revealed pentanucleotide motif GCAGC (Fig. 3e, Supplementary Fig. 6). On the other hand, modification loci in the LQ R20-0026 all centered on the motif GCTGG (Fig. 3f). Together, these results suggested that two lineage-specific modification systems extensively edited the five LQ genomes. Although their underlying mechanisms remained unclear, the editing at specific motifs with high conservation within each lineage allowed cost-effective in silico correction of these errors by Modpolish.

**Modpolish improved the quality of public ONT R9.4 datasets.** We then assessed the performance of Modpolish on public ONT datasets sequenced by R9.4 (SUP) and R10.4 flowcells (SUP, duplex/simplex modes). In the R9.4 dataset[14], we first compared the quality of seven bacterial genomes polished by Medaka and Modpolish (Fig. 4a, Supplementary Table 14). The quality of five genomes significantly improved from ~Q45 to Q60. Similarly, the

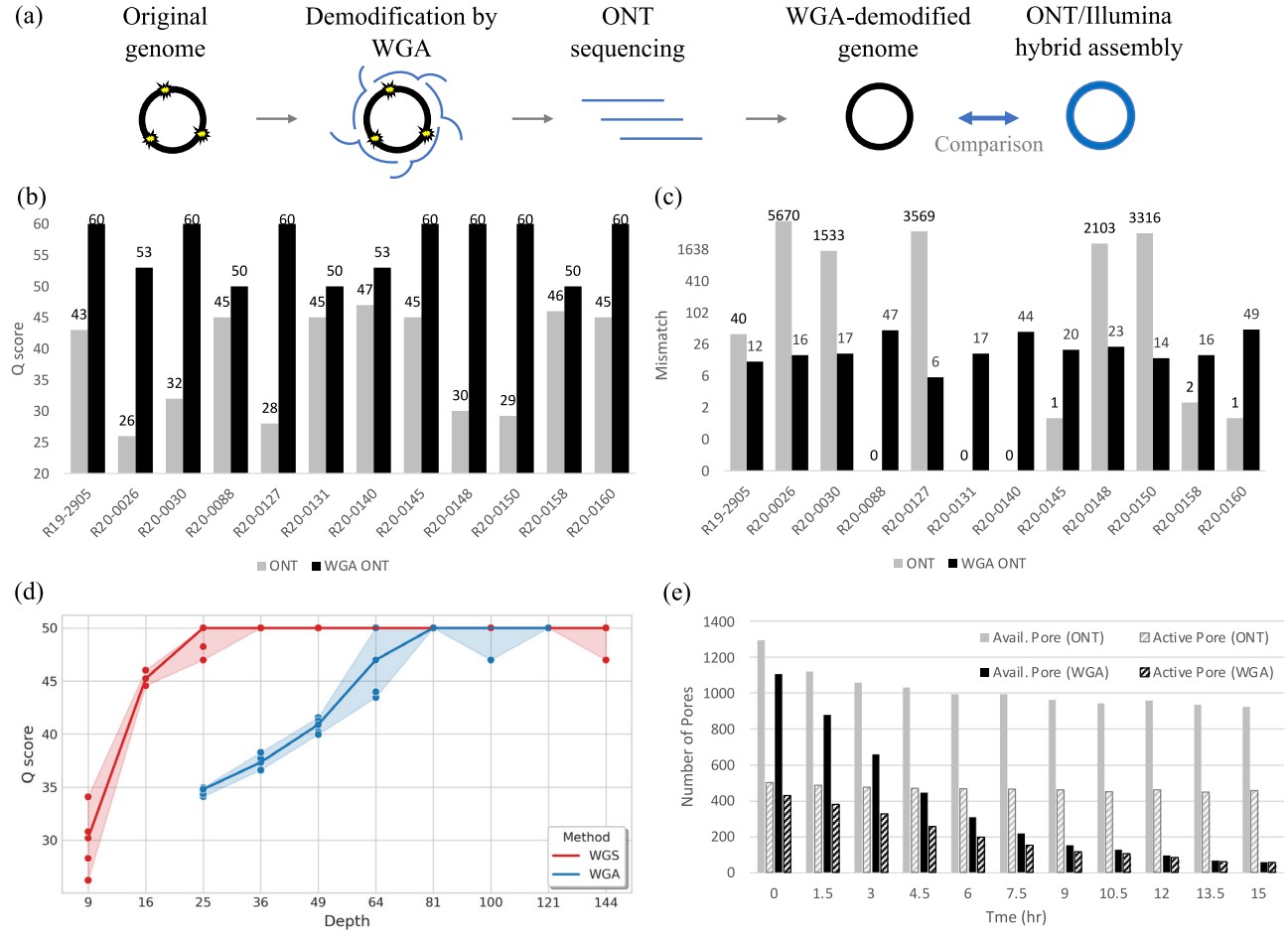

**Fig. 2 Quality improvement of ONT by WGA demodification. a** Worflow of WGA-demodified ONT; **b** Q scores of the WGA-demodified and ONT-only genomes (gray: ONT, black: WGA ONT); **c** numbers of mismatches of the WGA-demodified and ONT-only genomes (gray: ONT, black: WGA ONT); **d** WGA and ONT-only genome quality with respect to sequencing depth (shading: mininum and maximum quality in five replicates, line: median quality); **e** numbers of active/available pores during WGA-demodified and ordinary ONT sequencing.

improvement was mainly due to the reduction of mismatches (Fig. 4b). For instance, the number of mismatches decreased from 388 to 13 in the *Staphylococcus* genome after Modpolish. On average, the mismatch reduction rates of all genomes ranged from 50-96%. Consequently, although these bacterial genomes are not extensively modified, Modpolish can further improve their quality after Medaka without false corrections.

In the R10.4 (duplex mode) dataset[3], we compared the genome qualities polished by Medaka and Modpolish (downsampled to ~60×) (Fig. 4c, Supplementary Table 15). In general, Modpolish made little or no improvement in the duplex dataset. For instance, the mismatches produced by Modpolish only reduced from 20 to 19 on the *Bacillus* genome (Fig. 4d). The overall genome quality is very high such that no differences can be seen (Q60). Modpolish demonstrated marginal on a recently published simplex dataset (R10.4, kit 14, Dorado v0.1.1) (Supplementary Fig. 9). Therefore, the qualities of ONT R10.4 flowcells, in particular the duplex mode, is not only higher than those of R9.4 and require nearly no further correction. On the other hand, Modpolish may be used to fill the accuracy gap between simplex and duplex modes when the projects aim for higher throughput.

**Discussion**

This paper presented a set of unexpected low-quality ONT genomes due to extensive modifications untrained in the basecalling models. Demodification by WGA successfully improved the genome quality

while losing the epigenome. The in silico method, Modpolish, removed these modification-mediated errors without prior knowledge of modifications and uncovered the modified motifs while retaining the epigenome. When unknown modifications extensively shaped the genome, ONT with WGA or Modpolish produced nearly identical cgMLST profiles as hybrid ONT/Illumina did. Note that the hybrid ONT/Illumina assembly is not a perfect ground truth. On the other hand, the phylogeny of ONT-only genomes was disturbed by modification-mediated errors. Therefore, ONT with WGA or Modpolish is robust to modification-mediated errors without the need for additional Illumina sequencing.

Existing ONT basecalling algorithms only capture a few methylations (e.g., 5mC, 5hmc, 6 mA) and ignore the vast amount of other modifications. Theoretically, species-specific modifications can be distinguished by training bespoke models for one organism (e.g., Taiyaki). But practically, it is infeasible to train models for hundreds of modifications in the biosphere. Especially in metagenomic sequencing, the usage of any particular model is biased against other modifications. For instance, meta-epigenomic sequencing uncovered 22 methylation systems in a single microbial community[15]. Hence, these untrained modification-mediated errors are better removed by WGA demodification or Modpolish (viable only when large contigs can be obtained).

The cost of WGA ONT is higher than ordinary sequencing due to several side effects of the amplification. The WGA using

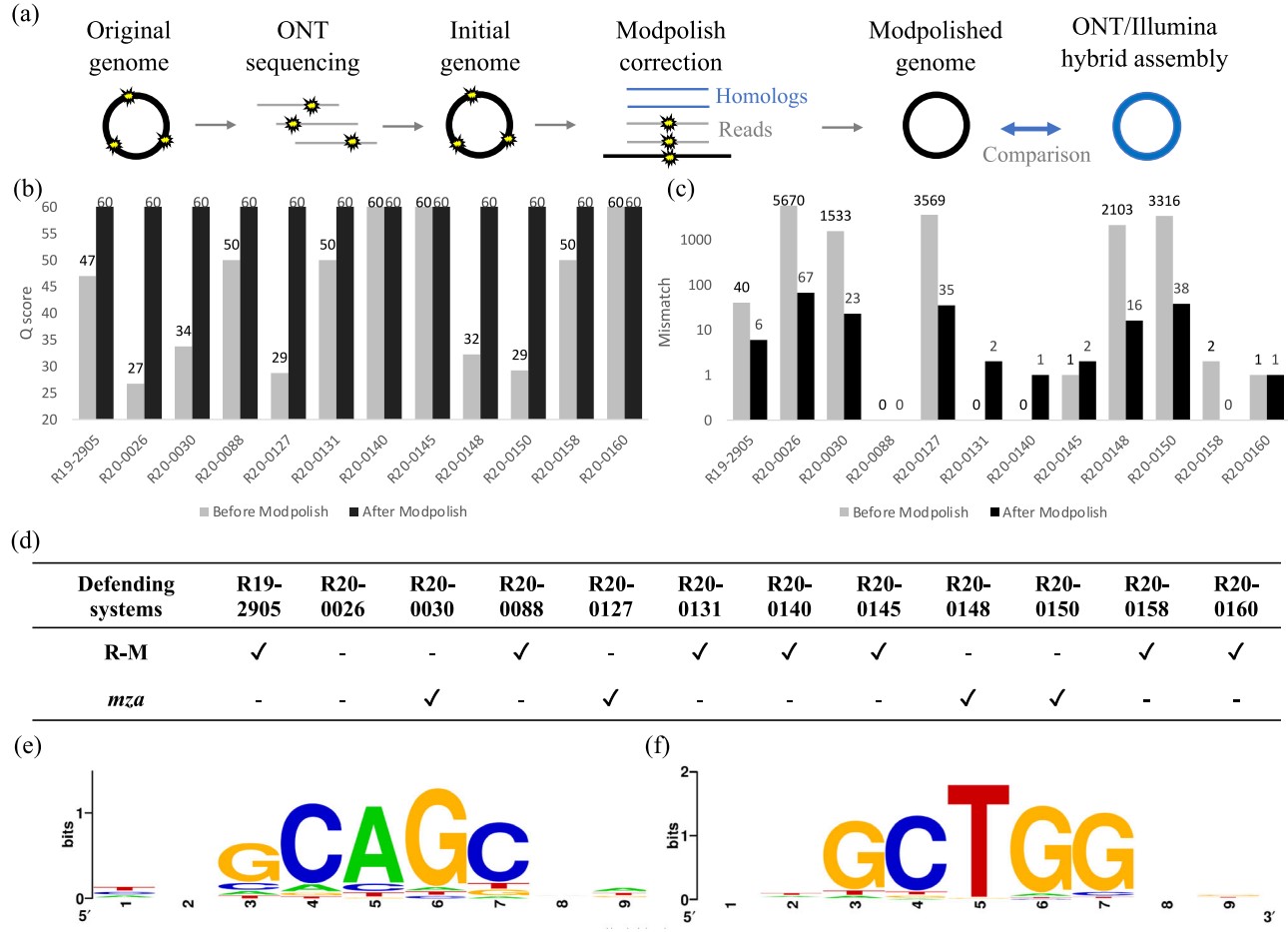

**Fig. 3 Correction of modification-mediated errors by Modpolish. a** Workflow of Modpolish; **b** Q scores before and after Modpolish; **c** numbers of mismatches before and after Modpolish (gray: before Modpolish, black: after Modpolish); **d** the antiviral defending systems encoded by the 12 strains (gray: before Modpolish, black: after Modpolish); **e** the sequence motif of modification sites in the four *mza*-encoding strains; **f** the sequence motif of modification sites on the R20-0026 strain.

multiple displacement amplification (MDA) can generate hyperbranched DNA structures that could block the pores and impede nanopore sequencing[10,16,17]. On the other hand, PCR-based WGA with adapter ligation might benefit the efficiency of nanopore sequencing but could have more mismatches due to a higher error rate at the amplification. Because MDA can generate much longer DNA with lower errors and better genome coverage than PCR does, this study chose MDA instead of PCR when implementing WGA.

Furthermore, we observed potentially amplification-mediated errors for WGA ONT than ordinary ONT in 5 of the 7 HQ strains (Fig. 2c). A higher amount of sequence reads (coverages) may be able to correct amplification errors for WGA-ONT as seen in the two remanding HQ strains. Finally, the WGS ONT generated longer contig lengths than WGA ONT for all strains but one (R20-0158) (Supplementary Table 16). Consequently, while WGA can erase these modification-mediated errors, the workflow complicates the sequencing, increases the turnaround time needed for clinically relevant answers, and removes the modifications that might be essential in the future.

While Modpolish eliminated most modification-mediated errors, the correction power was lower in the ST1081 isolate. The lack of ST1081 genomes in NCBI RefSeq decreased the sensitivity of Modpolish. As the algorithm only corrects the loci of high evolutionary conservation, a sufficient number of closely related genomes is necessary. The default parameters (e.g., number of related genomes and ANI cutoff) were optimized for

common species tested in this study. Therefore, Modpolish is more suitable for common instead of rare lineages at the default setting. While the user can fine-tune these parameters according to the abundance in the NCBI database, we note that this is a tradeoff between sensitivity and specificity.

Nevertheless, Modpolish retains all modifications after ONT sequencing while WGA loses the epigenome. In addition, this method is not limited to *Listeria* (e.g., quality improvement of *E. coli*, Supplementary Fig. 10), suggesting no prior knowledge on the underlying modification system is required. Epigenetic methylation has been thought to contribute to the rapid adaptation of resistance[18]. For instance, phase-variable adenine DNA methyl-transferases (e.g., ModA11 and ModA12) increase susceptibility to cloxacillin and ciprofloxacin in *Neisseria meningitidis*[19]. The resistance due to overexpression of efflux pumps (e.g., sugE) has been linked to the lack of Dcm-mediated 5mC silencing[20]. Therefore, Modpolish should be used when the epigenome is the focus of the study. Because extensive uncorrected errors can alter the distances between closely related strains, we assessed the reliability of core-genome MLST (cgMLST) phylogeny of the five LQ *Listeria* strains by four methods: ONT-only sequencing, WGA-demodified ONT, ONT with Modpolish, and hybrid ONT/Illumina sequencing (Supplementary Figure 11). The ONT-only genomes (polished by Medaka) were phylogenetically distant from the others due to excessive amounts of modification-mediated errors. On the other hand, the WGA-demodified genomes perfectly clustered with the ONT/Illumina hybrid for each strain in both clades (ST87 and

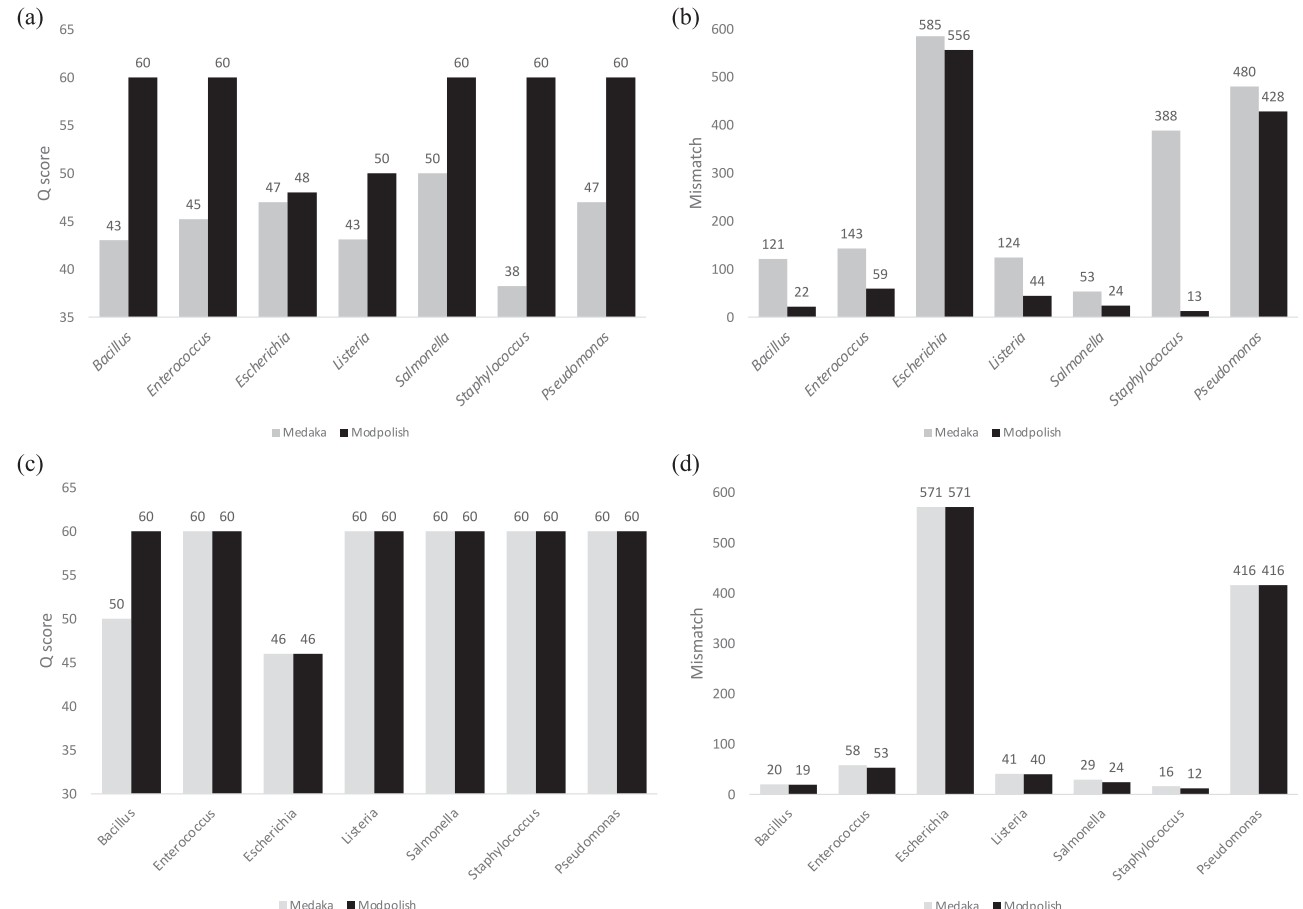

**Fig. 4 Evaluation of Modpolish on the Zymo ONT R9.4 (SUP) and R10.4 (Duplex, SUP) datasets.** Comparison of Medaka and Modpolish for **a** Q scores and **b** mismatches on the R9.4 dataset; comparison of Medaka and Modpolish for **c** Q scores and **d** mismatches on the R10.4 dataset.

ST1081). The ONT genomes corrected by Modpolish also clustered with the hybrid and WGA-demodified genomes in both clades, albeit the genetic distance slightly deviated in the ST1081 clade. Hence, we concluded the differences between WGA-deomodified and Modpolish genomes are marginal and can be ignored.

We discovered two pentanucleotide motifs, GCTGG (CCAGC) and GCAGC (GCTGC), specific to each of the two LQ lineages (ST1081 and ST87). In ST1081, the GCTGG (CCAGC) motif is part of *chi* sites, hotspots of homologous recombination mediated by the RecBC enzyme which degrades phages cut by restriction enzymes are further degraded by RecBC[21,22]. However, further studies are reuiqred to undertand the functional role of modifications on *chi* sites.

In ST87 strains, the GCAGC/GCTGC (i.e., GCWGC) motif was the known target of the orphan methyltransferase M.BatI[23]. M.BatI produced fully-methylation on 5′-GCWGC-3′ and hemi-methylation on 5′-GCSGC-3′. Reinvestigation of the modified sites in ST87 showed the existence of both GCWGC and GCSGC (Supplementary Figure 6). Interestingly, M.BatI increased toxicity when expressed in *E. coli* in their study, which was concordant with the elevated virulence of the four ST87 strains.

Hence, the two lineages possessed two distinct modification systems for defensive purposes and increasing virulence. Although further investigations are required to assess their biological function, modifications that have acquired regulatory effects in bacteria are usually conservative within a clade[24]. Consequently, our in silico algorithm successfully utilize the conservation for correcting modification errors.

This work reported a set of unexpectedly low-quality genomes due to novel modification types untrained in the ONT basecalling model. The increasing number of new modifications untrained in the basecalling models will unavoidably reduce the ONT accuracy. New ONT flowcells, sequencing kits, and basecalling algorithms will certainly resolve known modification-induced errors but not all of them. Our study showed that untrained modification-mediated errors could be effectively corrected by presequencing amplification or postassembly polishing without additional short-read sequencing, producing high-quality genomes reliable for downstream analysis.

## Methods

**Bacterial isolates**. Twelve *L. monocytogenes* isolates used in this study were obtained from hospitals recovered from listeriosis patients in Taiwan between 2019 and 2020. Written informed consent was obtained from all study participants. Ethical approval was granted by the participating hospitals in compliance with the Declaration of Helsinki principles. The isolates were submitted to the Taiwan Centers for Disease Control for further identification and genotyping. The isolates belonged to serogroups IIa (5 isolates), IIb (6 isolates), and IVb (1 isolate) and sequence type (ST) 1, ST5 (2 isolates), ST87 (4 isolates), ST101, ST155, ST378, ST1081, and ST1532.

**Whole genome sequencing**. WGS of bacterial isolates was conducted in the Central Region laboratory of Taiwan CDC using the

Illumina MiSeq sequencing platform (Illumina Co., USA) and the Nanopore sequencing platform (Oxford Nanopore Technologies, Inc., UK). DNA of bacterial isolates was extracted using the Qiagen DNeasy blood and tissue kit (Qiagen Co., Germany). Illumina DNA library construction was performed using the Illumina DNA Prep, (M) Tagmentation system (Illumina Co.), and sequencing was run with the MiSeq reagent kit version 3 ($2 \times 300$ bp), manipulated according to the manufacturer's instructions. Nanopore DNA library construction was performed using the Rapid Barcoding Kit and sequencing was run using the MinION device and R9.4 chemistry.

**Removal of modifications of nucleotides using whole-genome amplification**. DNA Bacterial Genomic DNA was amplified using the REPLI-g Advanced DNA Single Cell Kit (Qiagen, Hilden, Germany), and manipulated according to the manufacturer's instructions. The amplified DNA was purified using the KAPA HyperPure Beads (Roche, Basel, Switzerland) before being subjected to Nanopore sequencing.

**Assembly of sequence reads**. Illumina sequence reads for each isolate were assembled using the SPAdes assembler version 3.12.0 (http://cab.spbu.ru/software/spades/)[25]; both Illumina sequence reads and Nanopore sequence reads for each isolate together were assembled to complete the full genomic sequences using the Unicycler Assembler[26]. The Nanopore reads for each isolate (in the FAST5 file) were initially basecalled using Guppy v4.0.14 with the HAC model and later rebasecalled by v6.3.4 with the SUP model. In the ONT-only assembly, the sequences (in FASTQ file) were assembled using Flye (https://github.com/fenderglass/Flye)[27], then polished using the Racon (https://github.com/lbcb-sci/racon)[4], the Medaka (https://github.com/nanoporetech/medaka), and the Homopolish (https://github.com/ythuang0522/homopolish)[5]. Methylations (i.e., 5mC, 6 mA) in the ONT-only genomes were called by Megalodon (https://github.com/nanoporetech/megalodon). The Integrative Genome Viewer (IGV) was used for visualizing the ONT modification errors[28]. The genome quality was assessed by Fastmer (https://github.com/jts/assembly_accuracy).

**cgMLST analysis**. Assembled Illumina contigs, assembled and polished Nanopore contigs, and assembled complete genomic sequence (obtained from assembling Illumina sequences and Nanopore sequences) for each isolate were used to generate core-gene multilocus sequence typing (cgMLST) profiles (based on 2,172 core genes) using an in-house-developed cgMLST profiling tool available on the openCDCTW Github repo (https://github.com/openCDCTW/Benga). Phylogenetic trees were constructed with cgMLST profiles using the minimum spanning tree algorithm.

**Overview of Modpolish**. The proposed computational method, Modpolish, aims to remove modification-mediated errors by leveraging the inconsistency of basecalled nucleotides, qualities of basecalled alleles, and evolutionary conservation at the modified loci. Modpolish is an extension of Homopolish, a polishing algorithm designed for correcting ONT homopolymer errors[5]. Figure 5 depicts the workflow of Modpolish. The closely related genomes are first identified by screening against a compressed representation of microbial genomes in NCBI RefSeq. The genome sequences are then retrieved on the fly and compared with the draft genome. We only retain closely related genomes of high nucleotide and structural similarity. Given the alignment matrix of reads, qualities, and homologs, Modpolish identifies potential-modified loci of inconsistent basecalling and low quality and only

corrects the mismatch errors highly conserved in homologs. The details are described in the following sections.

**Collection of homologs by nucleotide and structural similarity**. The draft genome (to be polished) is scanned against the virus, bacteria, or fungus genomes compressed by Mash as (MinHash) sketches, which is a reduced representation of all microbial genomes in NCBI RefSeq[29]. Subsequently, top $t$ (default 20) closely related genomes (>95% Mash identity) will be retrieved on the fly. Mash estimated the Jaccard similarity between the draft and related genomes over a subset of sampled $k$-mers. Though very fast, this method has low resolution at distinguishing closely related genomes because the small subset of $k$-mers may not capture the few strain variations. Consequently, the genome similarity is further re-estimated using the more sensitive FastANI[30].

Each downloaded genome is further compared against the draft genome using FastANI for computing the average nucleotide identity (ANI) at a higher resolution than Mash. FastANI chops the two genomes into pieces and aligns them against each other for speedup. However, it only considers the aligned segments for ANI estimation and ignores the unaligned ones (Supplementary Fig. 12a). The unaligned segments imply these two genomes differ by large structural variations (i.e., vertically-/horizontally-transferred genes). As small and large variants are both genetic footprints of strain variations during evolution, Modpolish also estimates the structural similarity (average-structural identity, ASI), defined as the percentage of aligned segments. We only retain the related genomes with sufficient ANI (default: >99%) and ASI (default: >90%) for subsequent error correction. These empirical cutoffs were determined by investigating the distributions of ANI and ASI in real microbial genomes. In practice, the majority of species can improve quality solely by related genomes with 95% identity by Mash. The other two filters (i.e., 99% ANI by FastANI and 90% ASI) are only beneficial for some species/strains. The requirement of 99% ANI (by FastANI) and 90% ASI will be ignored when insufficient genomes are retained (default: <3), and the genomes (exceeding 95% identity by Mash) will be used for polishing.

**Correction of modification-mediated errors by reads and homologs**. These closely related genomes with sufficient sequence and structural similarity are aligned against the draft genome via minimap2 (with the asm5 option)[31]. The raw ONT reads are also mapped against the draft genome by minimap2 (with map-ont option). We extract the basecalled nucleotides, basecalling qualities, and homologous alleles from the alignments. The aligned homologs reads, and qualities are converted into a table of several summary statistics (Supplementary Fig. 12b).

The summary statistics include the allele counts of A, T, C, and G separately for homologs and ONT reads, ignoring the insertion and deletion gaps. We identify the potentially modified sites according to the allele discordancy and average quality (see also Supplementary Fig. 12b). The allele discordancy is the frequency of alternative alleles (i.e., non-major ones) at one locus. The average quality was computed by averaging the qscores from all read bases at the same locus. A potentially modified locus is defined as the allele discordancy greater than 5% and the average quality score below 15, which were empirically observed from the modification-mediated errors.

For each potentially modified locus, if all the homologous alleles are 100% conserved, we will correct the erroneous nucleotide into the alternative allele concordant with the homologs. These stringent criteria aimed for specificity instead of sensitivity, ensuring little or no false corrections would be made. The modified motifs were extracted according to the corrected loci outputted by Modpolish

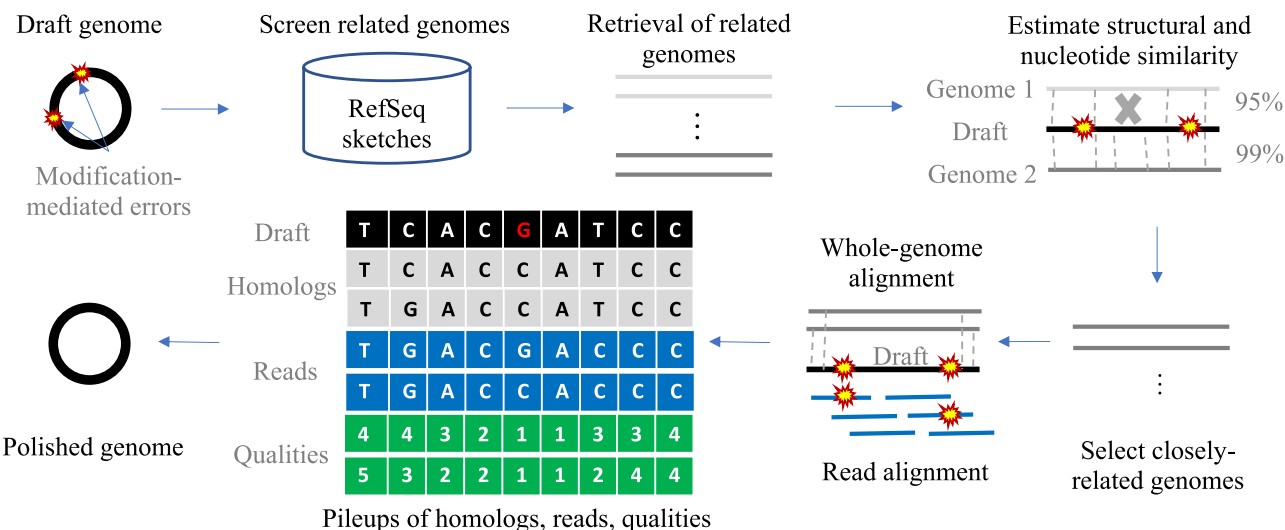

**Fig. 5 Illustration of Modpolish workflow.** A set of closely related genomes are first retrieved by screening the compressed sketches of RefSeq genomes. We retain the genomes with sufficient nucleotide and structural similarity. The selected genomes and ONT reads are aligned onto the draft genome, generating a pileup matrix of homologs, reads, and qualities. Modpolish only corrects modification-mediated errors with inconsistent read alleles, low quality, and high conservation in homologs.

with custom scripts. We implemented a motif-aware mode when the modification system is known in advance. If the user specifies a known modified motif (e.g., CCGAC), the program will additionally correct loci according to the provided pattern by lowering the homologous conservation ratio from 100% to 80%.

**Statistics and reproducibility**. The numbers of modified loci (i.e., 5mC and 6 mA) between HQ and LQ genomes and the number of mismatches (i.e., modification-mediated errors) between them are assessed by Student's *t* test. This study includes 12 *L. monocytogenes* and seven public bacteria isolates during experiments. The modification-mediated errors were confirmed by two biologically replicated ONT sequencing.

**Reporting summary**. Further information on research design is available in the Nature Portfolio Reporting Summary linked to this article.

## Data availability
The Illumina, O.N.T., and W.G.A. O.N.T. raw reads of the 12 isolates were deposited in the NCBI Short Read Archives (SRA) under the BioProject PRJNA839535 with SRA accession numbers: SRS1239568 and SRS13025957-SRS13025967. The source data used to plot Figs. 1b, c, e, 2b–e, 3b, c, and 4 can be found in Supplementary Data 2.

## Code availability
The Modpolish was implemented as a subcommand in the Homopolish package, which is freely available at (https://github.com/ythuang0522/homopolish/)[32].

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

## Acknowledgements

Y.T.H. was supported by the National Science and Technology Council, Taiwan (grant Nos. 111-2221-E-194-031-MY3 and 112-2628-E-194 -001-MY3). C.S.C. was supported by the Ministry of Health and Welfare, Taiwan (grant No. MOHW111-CDC-C-315-124306).

## Author contributions

C.S.C. and Y.T.H. conceived the study. B.H.C. and Y.W.W. performed the whole genome amplification, sequencing, and analysis. N.T.K., C.H.C., and Y.T.H. designed and implemented Modpolish. C.S.C. and Y.T.H. wrote the paper. All authors approved the paper.

## Competing interests

The authors declare no competing interests.
