## [Peer Review File · Communications Biology]

Reviewers' comments:

Reviewer #1 (Remarks to the Author):

I reviewed the paper as Reviewer #1 last time. Unfortunately, the authors did not directly address my main comment. I think it is of utmost importance that potential users can quickly gauge if this approach is applicable to their dataset, without having to read the supplementary of a previous paper. I should have formulated my concerns more explicitly. Hence, I'm adding it here again in a rephrased and more direct version that can be implemented by the authors (Major comment).

Major comment:

The claims in the abstract are not supported by the data presented in the paper. I am perplexed on why the authors do not want to be more transparent about their method. It has some great use-cases - but for many users, their data does not meet the limitations of Modpolish. This is completely normal for all software and needs to be transparently shown to potential users.

A) The title and abstract does not state that this is a reference-based method. If Modpolish were a de-novo method, it would be of great use to the vast majority of ONT R9.4 users. However, it is not. Please add this directly in the title and abstract. This is a large difference to fx. Racon and Medaka which the authors sometimes compare to in their review response.

B) The abstract does not mention that the method is only useful in the cases where a high number of very closely related genomes exists in the database. Basically, limiting the scope of Modpolish to clinical samples of common pathogens. Please add the recommended number of genomes and the identity cutoffs directly in the abstract (20 genomes within 99% ANI and 90% ASI, as far as I understood from the methods).

C) The abstract mentions: "Modpolish not only significantly improved the quality of in-house sequenced genomes but also public datasets sequenced by R9.4/R10.4 flowcells". However, from figure 4 it is only R9.4 where significant improvements are seen in Q-score. Results for R10.4 (which everyone hopefully is using now) show that there is no improvement or very limited. For example, estimated from figure 4, *Bacillus subtilis* improves from Q43 (medaka) to Q60 (modpolish), but for R10.4, medaka and Modpolish is both at Q42, and no improvement is seen for any of the genomes. As you show on the Zymo mock data, that the improvement from ONT R9.4 to R10.4, means that there is no advantage in Modpolish anymore I think it is essential to show if this is also the case for "novel" modifications on the *Listeria monocytogenes* genomes. Currently the only thing your data support is that Modpolish works for R9.4 - you have not shown it for R10.4. This needs to be very clear in the abstract.

Minor comments:

- The authors have made a sensitivity and specificity analysis based on my comments (Table S13-15). However, I likely did not make myself clear enough on what I thought was needed. I am concerned about the cases where few related organisms are present in the database and what that has of impact. Furthermore, what impact coverage has on the ability to correct and identify the errors? For table S13-15, I would like that for each species it is explicitly stated how many closely related genomes were used for correction and what the coverage was. The coverage is only broadly stated as 10x in Table S15 - which also seems strangely low.

- From the methods, it is a little unclear what the cutoff in genomes used for correction is. Methods state that the top-20 closest related genomes are taken. However, it seems like 20 was before

ANI+ASI cutoffs? In my mind, the correct cutoff has to be defined pr. position as some positions might not be present in all closely related genomes?

- Throughout the manuscript, please add which ONT version was used directly in the text. I.e. R9.4 or R10.4.

- From the methods and review response, it is mentioned that both HAC and SUP base-calling was used. I could not see from the figures if HAC or SUP was used. Please make sure that all your analysis only uses SUP. It adds confusion and makes no sense to compare Modpolish with HAC.

- I do not think the data supports your claim that modpolish significantly improves the accuracy over R10.4 (line 154-166) to me it looks the same at 50x? As sequencing cost is such a small part of the total price to sequence a genome I do not think that anyone would generally go for less than 100x anyway for isolate sequencing. We usually get 1000x+++ because we can not multiplex enough genomes. Please re-analyse the data and rephrase the section (also relates to Major Comment part C).

- Figure 4: It is very difficult to compare the R9.4 and R10.4 data. Please integrate the R9.4 and R10.4 into results into 1 figure (Q score and mismatches separately). Furthermore, if the Q-score bar-plot is chosen, please compare using 50x coverage for all datasets so the data can be compared (I could not see what coverage was used for the R9.4 data currently?). Optimally I would like to see the coverage profile as shown for the R10.4 data for both R9.4 and R10.4 combined. Finally, do not show the flye error rate as it is not essential for the comparison (also relates to Major Comment part C).

- Line 184-188: I do not see Modpolish currently being useful in a metagenome context, as the vast majority of genomes recovered from metagenomes will have no or very few closely related genomes in the database. This is by far the largest challenge.

- Line 220-228: Add the limitations directly to the text. What were the default settings you implemented? It is difficult to read this from the methods(number of genomes, ANI, ASI, % consensus).

Limitations to this review:

Unfortunately, I did not have sufficient time to test the software itself but I hope other reviewers had, as it is an essential part of the review. Furthermore, I'm not an expert in restriction systems and can not evaluate the scientific findings in this regard.

Reviewer #2 (Remarks to the Author):

I think the authors have adequately addressed most reviewers' comments and the manuscript have improved. But some minor points raised below would need to be addressed.

Line 201 and 204: I think it is more common to express errors caused by DNA amplification as "amplification errors", rather than "polymerization errors".

Line 211-216: References supporting the descriptions should be cited.

Correcting Modification-Mediated Errors in Nanopore Sequencing by Nucleotide Demodification and Reference-Based Correction (Revision Report)

General Response:

Dear editor and reviewers,

We have revised the manuscript by following Reviewer #1's suggestions (i.e., emphasizing the usage scope of Modpolish to R9.4, update of all numbers to latest SUP model, ..etc). The remaining comments are centered on the explicitly stating the limitations of the program, which we agree. However, we hope the editor and reviewers could also see this work highlights the merits of nucleotide demodification by whole-genome amplification and the novel modification system in *Listeria* untrained in the Nanopore basecalling model. Below please find point-to-point responses to reviewer #1's comments. We hope the editor and reviewers could accept the manuscript for publication.

Reviewer #1 (Remarks to the Author):

Major comments:

A) The title and abstract does not state that this is a reference-based method. If Modpolish were a de-novo method, it would be of great use to the vast majority of ONT R9.4 users. However, it is not. Please add this directly in the title and abstract. This is a large difference to fx. Racon and Medaka which the authors sometimes compare to in their review response.

Ans: The title has been changed to "Correcting Modification-Mediated Errors in Nanopore Sequencing by Nucleotide Demodification and Reference-Based Correction" in the revised manuscript. We have also revised the abstract by adding reference-based correction in the corresponding sentences.

B) The abstract does not mention that the method is only useful in the cases where a high number of very closely related genomes exists in the database. Basically, limiting the scope of Modpolish to clinical samples of common pathogens. Please add the recommended number of genomes and the identity cutoffs directly in the abstract (20 genomes within 99% ANI and 90% ASI, as far as I understood from the methods).

Ans: We have revised the sentences in the abstract. "We developed a reference-based method, Modpolish, for correcting modification-mediated errors while retaining the epigenome when a sufficient number of closely-related genomes is publicly available (default: 20 genomes with 95% ANI)". The reviewer's original suggestion would be misleading as a hierarchical similarity estimation was implemented, i.e., 95% ANI by Mash first, 99% ANI by FastANI, and finally 90% ASI. In practice, the majority of species can improve quality solely by related genomes with 95% ANI by Mash. The other two filters (i.e., 99% ANI by FastANI and 90% ASI) are only beneficial for some species/strains. The requirement of 99% ANI by FastANI and 90% ASI will be ignored when insufficient genomes are retained, and all

genomes exceeding 95% ANI by Mash will be used. We found it's difficult to explain this hierarchical filtration in the abstract, and hope the reviewer can accept this shorter version which stands for most cases. We have also revised the method for clarifying this hierarchical filtration (L355-360).

C) The abstract mentions: "Modpolish not only significantly improved the quality of in-house sequenced genomes but also public datasets sequenced by R9.4/R10.4 flowcells". However, from figure 4 it is only R9.4 where significant improvements are seen in Q-score. Results for R10.4 (which everyone hopefully is using now) show that there is no improvement or very limited. For example, estimated from figure 4, Bascillus subtilis improves from Q43 (medaka) to Q60 (modpolish), but for R10.4, medaka and Modpolish is both at Q42, and no improvement is seen for any of the genomes. As you show on the Zymo mock data, that the improvement from ONT R9.4 to R10.4, means that there is no advantage in Modpolish anymore I think it is essential to show if this is also the case for "novel" modifications on the Listeria monocytogenes genomes. Currently the only thing your data support is that Modpolish works for R9.4 - you have not shown it for R10.4. This needs to be very clear in the abstract.

Ans: We have followed the reviewer's suggestion by emphasizing the improvement on R9.4 only in the abstract.

Minor comments:

- The authors have made a sensitivity and specificity analysis based on my comments (Table S13-15). However, I likely did not make myself clear enough on what I thought was needed. I am concerned about the cases where few related organisms are present in the database and what that has of impact. Furthermore, what impact coverage has on the ability to correct and identify the errors? For table S13-15, I would like that for each species it is explicitly stated how many closely related genomes were used for correction and what the coverage was. The coverage is only broadly stated as 10x in Table S15 - which also seems strangely low.

Ans: We have disclosed the limitations of common species in the abstract of this revision.

“We developed a reference-based method, Modpolish, for correcting modification-mediated errors while retaining the epigenome when a sufficient number of closely-related genomes is publicly available.”

Second, all the genomes tested in this manuscript (Tables S13-15) are common species and thus can retrieve the default 20 related genomes (>95 ANI by Mash). As we have explicitly sequenced and evaluated over these rare bacteria in the previous publication to justify this limitation (Huang et al., 2020), we hesitate to do this again as not much improvement will be expected (Please also refer to response #1 in the previous round of revision). We hope the reviewer could understand it's extremely difficult to culture some of the rare species (e.g., anaerobic microbes).

The coverage was added in Supplementary Table S13 according to the sequencing and assembly stats in Supplementary Tables S2 and S4.

Supplementary Table S13. The coverage, true positive (TP), true negative (TN), false positive (FP), false negative (FN), sensitivity, and specificity of Modpolish corrections on the 12 *Listeria* strains.

	Coverage	TP	TN	FP	FN	Sensitivity	Specificity
R19-2905	121	43	3121746	0	1	97.73%	100.00%
R20-0026	110	5226	2936114	0	51	99.03%	100.00%
R20-0030	145	1122	2952319	5	10	99.12%	100.00%
R20-0088	183	8	3008962	0	0	100.00%	100.00%
R20-0127	221	3374	2989449	0	30	99.12%	100.00%
R20-0131	252	1	2952987	3	0	100.00%	100.00%
R20-0140	191	1	2927143	1	0	100.00%	100.00%
R20-0145	149	2	2895648	1	1	66.67%	100.00%
R20-0148	218	2698	2909920	0	15	99.45%	100.00%
R20-0150	147	3159	3011849	0	31	99.03%	100.00%
R20-0158	135	4	2945193	0	0	100.00%	100.00%
R20-0160	113	7	2986861	0	0	100.00%	100.00%

The two public datasets, Supplementary Tables 14-15, are metagenomic sequencing downloaded from the ZymoBIOMICS Microbial Community Standard, and thus difficult to assess the individual microbial coverage in the mixed community. We hope the reviewer can leave this part unspecified.

- From the methods, it is a little unclear what the cutoff in genomes used for correction is. Methods state that the top-20 closest related genomes are taken. However, it seems like 20 was before ANI+ASI cutoffs? In my mind, the correct cutoff has to be defined pr. position as some positions might not be present in all closely related genomes?

Ans: Please refer to response to comment #1. A multi-level similarity estimation was implemented, i.e., 95% ANI by Mash first, 99% ANI by FastANI, and finally 90% ASI. The 20 genomes refer to the 95% ANI by Mash.

The number of genomes may vary at different loci due to structural variations (SVs) (e.g., transposon-like elements ISs). In practice, it's hard to optimize this parameter at per-base resolution as the distribution of SVs also varies in the related genomes. We have tested this parameter at the early development stage (from 10 to 30) at the whole-genome scale and found ~20 can roughly handle the common and rare species. In fact, this default number (20) is much higher than needed as only a few (true) closely-related strains are sufficient for corrections in practice.

- Throughout the manuscript, please add which ONT version was used directly in the text. I.e. R9.4 or R10.4.

Ans: Done.

- From the methods and review response, it is mentioned that both HAC and SUP base-calling was used. I could not see from the figures if HAC or SUP was used. Please make sure that all your analysis only uses SUP. It adds confusion and makes no sense to compare Modpolish with HAC.

Ans: We have updated all the numbers to the SUP model (sequenced by ourselves) in the revised manuscript.

- I do not think the data supports your claim that modpolish significantly improves the accuracy over R10.4 (line 154-166) to me it looks the same at 50x? As sequencing cost is such a small part of the total price to sequence a genome I do not think that anyone would generally go for less than 100x anyway for isolate sequencing. We usually get 1000x+++ because we can not multiplex enough genomes. Please re-analyse the data and rephrase the section (also relates to Major Comment part C).

Ans: We are sorry for not clarifying simple vs duplex modes. We have revised the descriptions of R10.4 (duplex mode) to no improvement in the revised manuscript (Figure 4(c)(d)). In the simplex R10.4 dataset, marginal improvement can still be seen (Supplementary Figure S11).

- Figure 4: It is very difficult to compare the R9.4 and R10.4 data. Please integrate the R9.4 and R10.4 into results into 1 figure (Q score and mismatches separately). Furthermore, if the Q-score bar-plot is chosen, please compare using 50x coverage for all datasets so the data can be compared (I could not see what coverage was used for the R9.4 data currently?). Optimally I would like to see the coverage profile as shown for the R10.4 data for both R9.4 and R10.4 combined. Finally, do not show the flye error rate as it is not essential for the comparison (also relates to Major Comment part C).

Ans: We have updated Figure 4 as suggested. We have clarified the sentences to indicate no improvement on duplex and marginal improvement can be observed in the simplex mode.

“Therefore, the qualities of ONT R10.4 flowcells, in particular the duplex mode, is not only higher than those of R9.4 and require nearly no further correction by Modpolish. In the simplex mode, marginal improvement can be seen.”

Figure 4. Evaluation of Modpolish on the Zymo ONT R9.4 (SUP) and R10.4 (Duplex, SUP) datasets. Comparison of Medaka and Modpolish for (a) Q scores and (b) mismatches on the R9.4 dataset; comparison of Medaka and Modpolish for (c) Q scores and (d) mismatches on the R10.4 duplex dataset.

Supplementary Figure S11. Comparison of mismatches and Q scores of Medaka and Modpolish using the R10.4 simplex dataset (kit 14, Dorado basecaller v0.1.1, SUP).

- Line 184-188: I do not see Modpolish currently being useful in a metagenome context, as the vast majority of genomes recovered from metagenomes will have no or very few closely related genomes in the database. This is by far the largest challenge.

Ans: We have rephrased the sentence to convey the intended meaning.

“Hence, these untrained modification-mediated errors are better removed by WGA demodification or Modpolish (viable only when large contigs can be obtained).” We fully understood the limitations of Modpolish and would like to note that WGA demodification is also another option suggested in the manuscript.

- *Line 220-228: Add the limitations directly to the text. What were the default settings you implemented? It is difficult to read this from the methods(number of genomes, ANI, ASI, % consensus).*

Ans: See response to comment #1. We have emphasized these limitations in the abstract.

Response to Reviewer 2

I think the authors have adequately addressed most reviewers' comments and the manuscript have improved. But some minor points raised below would need to be addressed.

Line 201 and 204: I think it is more common to express errors caused by DNA amplification as “amplification errors”, rather than “polymerization errors”.

Ans: Done.

Line 211–216: References supporting the descriptions should be cited.

Ans: Done.

REVIEWERS' COMMENTS:

Reviewer #1 (Remarks to the Author):

I've previously reviewed the paper. I think the authors have made great progress in making it much more transparent in which cases modpolish could be applied. I only have a few comments left.

1. I recommend that a text professional go through the manuscript in detail and that the authors carefully look at the new modifications to the text. They sometimes conflict with previous statements. I've highlighted a few examples below from the start and the end of the paper:

Line 19: I don't think there is something called "newborn diseases".

Line 27-35: First you say that post-assembly polishing is compulsory, then that it is not?

Line 40-46: You say unfixable systematic errors, but then that many have been fixed already?

line47: There is a "=" sign in the manuscript?

line 270: "The increasing number of new modifications found by single-molecular sequencing or high-resolution mass spectrometry will unavoidably reduce the ONT accuracy." This makes little sense, the signal does not change just because a new modification is found. Rather it presents the opportunity to improve the ONT accuracy by improving the basecalling models (as you also state in the next sentence).

Line 275: You write pre-assembly amplification, it sounds a little strange. I would say that the important step is the sequencing. Hence pre-sequencing amplification if you want to introduce a specific term for it.

2. In the abstract I think you generalize your findings too much. I see your main contribution as a method to fix/mask modification-mediated errors not currently in the ONT base-calling models. However in lines 11-12 you generalize your findings to public datasets sequenced by R9.4. I think it would be important to also mention the results with R10.4 simplex and duplex if you want to generalize your findings (as you do in the main section of the paper). I hope everyone has switched to R10.4 a long time ago. Hence, that is the most relevant pore type to compare with in order to generalize the findings.

3. One of your important arguments for correcting errors and making perfect genomes is that it is needed for cgMLST in clinical settings (where Nanopore has a clear time advantage). However, as far as I read, you first mentioned that in the discussion. In my opinion, you need to have some data presented in the results to discuss them afterward. Furthermore, instead of correcting the errors, wouldn't you get the same results if you just masked the positions with a Q-score filter? It's many years since I've played with MLST, but I think position-based quality filters should be a possibility in most software already.

4. If you still have the "low quality" listeria samples I would recommend sequencing them with R10.4. Showing that the novel modification-related errors remain there (as I would suspect) would be a powerful argument for your paper.

5. Your section on WGA/MDA is quite long compared to the insights it provides and is mainly based on your findings. MDA/WGA has been extensively used for years and multiple protocols have been published with tweaks to improve it. Hence, I feel your conclusions on the very very poor performance of an MDA-based workflow are premature and not generalizable in their current form. Hence, I would recommend cutting this section shorter and using the main argument that a WGA-based workflow complicates the sequencing, increases the turnaround time needed for clinically relevant answers, and

of course, removes the modification information that might be essential in the future. The section and claims on WGA and performance need more references as only 1 reference is mentioned in lines 188-215 regarding WGA. A few top-of-mind examples that can be used as starting points for relevant literature: <https://onlinelibrary.wiley.com/doi/full/10.1002/mco2.116>,
<https://www.nature.com/articles/s41592-021-01109-3>,
<https://onlinelibrary.wiley.com/doi/full/10.1111/mec.16679>.

Correcting Modification-Mediated Errors in Nanopore Sequencing by Nucleotide Demodification and Referense-Based Correction (Revision Report)

Point-to-point responses (Reviewer 1)

1. I reccommed that a text professional go through the manuscript in detail and that the authors carefully look at the new modifications to the text. They sometimes conflict with previous statements. I've highlighted a few examples below from the start and the end of the paper:

Line 19: I don't think there is something called "newborn diseases".

Line 27-35: First you say that post-assembly polishing is compulsory, then that it is not?

Line 40-46: You say unfixable systematic errors, but then that many have been fixed already?

line47: There is a "=" sign in the manuscript?

line 270: "The increasing number of new modifications found by single-molecular sequencing or high-resolution mass spectrometry will unavoidably reduce the ONT accuracy." This makes little sense, the signal does not change just because a new modification is found. Rather it presents the opportunity to improve the ONT accuracy by improving the basecalling models (as you also state in the next sentence).

Line 275: You write pre-assembly amplification, it sounds a little strange. I would say that the important step is the sequencing. Hence pre-sequencing amplification if you want to introduce a specific term for it.

Ans: Thanks. We have corrected these errors and proofread again.

2. In the abstract I think you generalize your findings too much. I see your main contribution as a method to fix/mask modification-mediated errors not currently in the ONT base-calling models. However in lines 11-12 you generalize your findings to public datasets sequenced by R9.4. I think it would be important to also mention the results with R10.4 simplex and duplex if you want to generalize your findings (as you do in the main section of the paper). I hope everyone has switched to R10.4 a long time ago. Hence, that is the most relevant pore type to compare with in order to generalize the findings.

Ans: We have revised the abstract accordingly by adding the R10.4 simplex/duplex results.

3. One of your important arguments for correcting errors and making perfect genomes is that it is needed for cgMLST in clinical settings (where Nanopore has a clear time advantage). However, as far as I read, you first mentioned that in the discussion. In my opinion, you need to have some data presented in the results to discuss them afterward. Furthermore, instead of correcting the errors, wouldn't you get the same results if you just masked the positions with a Q-score filter? It's many years since I've played with MLST, but I think position-based quality filters should be a possibility in most software already.

Ans: The cgMLST is briefly mentioned in the discussion as this is only one of several benefits of correcting modification-mediated errors (e.g., motif analysis). The quality is one of the features used for identifying modifications. By simply masking all low-quality bases of the genomes, one will get false positives (e.g., unmodified sites due to aberrant signals) and false negatives (e.g., we see some modified sites of high qualities). Therefore, we developed Modpolish which considers not only qualities but also read and homolog conservation.

4. If you still have the "low quality" listeria samples I would recommend sequencing them with R10.4. Showing that the novel modification-related errors remain there (as I would suspect) would be a powerful argument for your paper.

Ans: This will be the future work as R10.4 was delayed in Taiwan.

5. Your section on WGA/MDA is quite long compared to the insights it provides and is mainly based on your findings. MDA/WGA has been extensively used for years and multiple protocols have been published with tweaks to improve it. Hence, I feel your conclusions on the very very poor performance of an MDA-based workflow are premature and not generalizable in their current form. Hence, I would recommend cutting this section shorter and using the main argument that a WGA-based workflow complicates the sequencing, increases the turnaround time needed for clinically relevant answers, and of course, removes the modification information that might be essential in the future. The section and claims on WGA and performance need more references as only 1 reference is mentioned in lines 188-215 regarding WGA. A few top-of-mind examples that can be used as starting points for relevant literature: <https://onlinelibrary.wiley.com/doi/full/10.1002/mco2.116>, <https://www.nature.com/articles/s41592-021-01109-3>, <https://onlinelibrary.wiley.com/doi/full/10.1111/mec.16679>.

Ans: We have shorten the discussion section of WGA/MDA and added more references as requested.